# Clinical Decision-Making for Appendectomy in Kosovo: A Conjoint Analysis

**DOI:** 10.3390/ijerph192114027

**Published:** 2022-10-28

**Authors:** Ilir Hoxha, Bajram Duraj, Shefki Xharra, Afrim Avdaj, Valon Beqiri, Krenare Grezda, Erza Selmani, Blerta Avdiu, Jakob Cegllar, Dorjan Marušič, Aferdita Osmani

**Affiliations:** 1The Dartmouth Institute for Health Policy and Clinical Practice, Geisel School of Medicine at Dartmouth, Lebanon, NH 03766, USA; 2Evidence Synthesis Group, 10000 Prishtina, Kosovo; 3Research Unit, Heimerer College, 10000 Prishtina, Kosovo; 4Lux Development, 10000 Prishtina, Kosovo; 5General Hospital of Prizren, 20000 Prizren, Kosovo; 6General Hospital of Gjilan, 60000 Gjilan, Kosovo; 7The Faculty of Medicine, University of Rijeka, 51000 Rijeka, Croatia

**Keywords:** appendectomy, preoperative care, conjoint analysis, low and middle income, healthcare reform, evidence-based practice

## Abstract

**Objective**: The objective was to investigate the association of clinical attributes with decision making for performing appendectomy and making preoperative preparations for appendectomy. **Method:** A conjoint analysis with 17 clinical scenarios was executed with surgeons employed at public hospitals in Kosovo. **Setting:** The study was conducted at two public hospitals in Kosovo that have benefited from quality-improvement interventions. **Participants:** The participants included 22 surgeons. **Outcome measures:** The primary outcome was the overall effect of clinical attributes on the decision to perform appendectomy and make the preoperative preparations for appendectomy. **Results**: In the regression analyses, several attributes demonstrated statistically significant effects on the clinical decision to perform appendectomy and on the practice of preoperative preparation. **Conclusions:** We found that several factors influenced the decision to perform appendectomy and the practices for preoperative preparation. Nevertheless, the small sample size limited our efforts to interpret the results. These findings could assist Kosovo in the design and implementation of future similar studies and in fostering quality improvement measures that address clinical decision making and the lack of process standardization in the delivery of surgical care.

## 1. Introduction

Appendicitis, as one of the most common causes for surgical intervention, remains a notable public health problem, with the highest incidence occurring in the second and third decades of life [1]. The appendix, a small organ located at the base of cecum, is considered a medical oddity without a confirmed role and with the potential of getting inflamed. The lifetime risk of having appendicitis has been estimated to range from 8.6% to 12% in males and 6.7% to 23.1% in females [1,2,3]. Only half of patients have a typical presentation, and some cases may have a delayed presentation. An atypical presentation is more common in very young, elderly, pregnant, or immunosuppressed patients [4]. Therefore, a clinical diagnosis is often challenging and includes a combination of clinical signs and symptoms, physical examination, laboratory results, and radiological findings [5]. Appendectomy is the standard surgical treatment procedure, and it is generally well-tolerated by patients, but it can be associated with postoperative morbidity [6,7]. Accordingly, the decision to perform an appendectomy on a patient with suspected appendicitis should be considered under a careful examination of various clinical factors and the outcome of previous similar cases. The surgeon should also consider other parameters such as the patient’s age and comorbidities, the diagnostic accuracy, and patient consent [8]. Despite all the diagnostic information available, the decision to operate or not remains a challenge [5].

Phase two of the Project KSV/017 Health Support Programme in Kosovo aimed to establish a quality and safety management system at two public general hospitals in Kosovo: Prizren and Gjilan Hospital. The process started with a situation analysis of the hospitals in terms of their service delivery (availability of services, accessibility, utilization, infrastructure, management, quality, and safety, as well as efficiency), human resources, Health Information System (HIS), availability of medication, equipment, financing, and leadership [9]. The assessment noted a major variation in clinical decision making and the clinical knowledge of surgical procedures such as appendectomies. The project continued with extensive effort to improve the provision of health services in the targeted hospitals. The central part of the support was the development of new standards of care via standard operating procedures (SOPs) as well as extensive training and knowledge transfer among the staff of both hospitals, which had lasted a little over a year by the time this study was implemented. The project supported the development of the SOP for the preoperative preparation of patients for appendectomy according to international standards, which Kosovo University Clinical and Hospital Service approved. It also provided medical staff with training for and knowledge of the procedure and the use of SOPs in clinical practice. As a result, we were interested in documenting the variation in clinical decision making and assessing the early impact of training (i.e., intervention) in changing the situation.

To achieve this, i.e., assess clinical decision making and the influence of training in changing decision-making patterns, we opted for a conjoint analysis. Conjoint analyses have previously been used to examine the importance of clinical factors in clinical judgments [10,11]. It is used to quantify preferences by performing discrete choice designs through multiple scenarios [12]. The conjoint analysis has recently become a popular analytical technique in healthcare [13,14] that resembles individuals’ daily decision making when choosing between multi-attribute alternatives and their levels of characteristics [15]. This approach offers facilitated decision making for performing a clinical procedure, such as appendectomy. It has the potential to assess clinical decision making, particularly medical practice variation. Variation in clinical decision making is an important source of problems in the delivery of care, where inappropriate care is over-provided and needed care is under-provided [16,17,18,19]. Studying this at the micro level, that is, the physician level, is extremely useful as compared to studying it at the regional level, because it provides direct insight into what is happening in clinical decision making. This makes it easier to understand the causes of variation and could be more helpful in changing existing clinical practice patterns, which could then lead to a standardization of care.

The objective of this study was to elicit the process of clinical decision making related to performing appendectomy and the preoperative preparation for appendectomy by evaluating surgeons’ preferences through discrete choice designs of various clinical scenarios. It aimed to document the variation in clinical decision making and the attributes that drive such variation, and to examine the early impact of training programs for surgeons.

## 2. Materials and Methods

Two conjoint experiments were performed. The first experiment focused on the clinical decision to conclude that an appendectomy should be performed based on symptoms and a clinical examination. The second focused on the preoperative preparation of patients for an appendectomy based on a clinical examination and other preparation procedures. A 10-stage structure of a conjoint analysis in health, adapted by Bridges et al. [12], was followed.

### 2.1. Study Sample and Data Collection

The participants were surgeons from two regional general hospitals in Kosovo. Both hospitals are beneficiaries of the KSV017 project. In total, 22 interviews were performed and only 2 eligible participants could not participate in the experiments. Even though larger sample sizes are recommended for performing a conjoint analysis, smaller population sizes may be employed when the population is representative of the source population [13,20,21,22].

The data collection was performed by an experienced researcher (I.H.) using in-person interviews in most cases. When that was not possible due to COVID-19, we switched to online interviews. Each respondent was asked to make a decision for each vignette. Each presented vignette/scenario had the options of yes or no. In the first experiment, the clinician had to decide if he would perform an appendectomy on a patient with acute appendicitis based on clinical signs and examination results. In the second experiment, the clinician had to decide whether the patient was (preoperatively) prepared to enter the operation room for an appendectomy procedure.

### 2.2. Attributes and Experimental Design

We performed a literature review of textbooks and research papers to determine the attributes and levels of attributes for both experiments. For the second experiment, we also referred to an SOP for the preoperative preparation of patients for appendectomy, which was developed by the project to assist the surgeon’s work concerning appendectomies. Attributes were selected based on three criteria: relevance to the research question, relevance to the decision context, and relation to one another [23]. As a result, we developed two sets of attributes and levels for each experiment. The number of attributes for each planned questionnaire was seven, with two to three respective levels (Table 1 and Table 2).

To test the attributes with clinicians, we compiled two questionnaires for each experiment. We used these tests to examine the clarity of the attributes and to identify if there were any perceived absolute indications to remove them after the testing. The attributes and respective levels were reviewed to produce a final list of attributes for both experiments. The attributes for the first experiment included the McBurney sign, vomiting, temperature, white blood cell count (WBCC), erythrocyte sedimentation rate (ESR), C-reactive protein (CRP), and other causes (not appendicitis-related) of right lower quadrant abdominal pain. The attributes for the second experiment were a clinical examination, clinical measures, comorbidities, diagnostic procedures, preoperative techniques, other procedures, and patient consent.

The next step was the design of the vignettes, i.e., clinical scenarios for each experiment. This was accomplished using the SPSS orthogonal design facility. As a result, we identified 16 vignettes for each planned questionnaire that contained different levels out of seven attributes. We added an additional vignette with a real-case scenario for both experiments. 

### 2.3. Measurements and Statistical Analysis

The vignettes for each experiment were integrated into the main survey questionnaire. In addition to the vignettes, these questionnaires collected information on demographics, education, experience, exposure to scientific work, and benefits from the project (i.e., intervention). Multinomial logistic regression models with robust variance estimates, yielding relative risk ratios (RRRs), reflected the probability of choosing to perform an appendectomy (Experiment 1) or deciding that the patient is prepared to have an appendectomy performed (Experiment 2). Statistical analyses were performed with STATA V17 BE (Stata Corp, College Station, TX, USA). The Ethical Review Committee reviewed the study protocol at Heimerer College. In the original design, we aimed to perform a subgroup analysis by project benefit, but due to the small sample size, we refrained from performing it.

## 3. Results

### 3.1. Participant Characteristics

The final sample consisted of 22 surgeons, including paediatric surgeons (18%) and general surgeons (82%). Only one respondent reported having a sub-speciality. A total of 21 respondents reported regular Continual Medical Education (CME) attendance. A considerable percentage of the participants had benefited from the training (73%), but only a small percentage (9%) had benefited from the study visits. None of the respondents claimed that their ward had benefited from the project.

### 3.2. Number of Vignettes and Cases 

A total of 17 vignettes were employed in each experiment. With 22 participants, there were 374 cases available for each analysis, i.e., the decision for appendectomy and preparation for appendectomy.

### 3.3. Determinants of Clinical Decision for Appendectomy

In the regression analyses, several attributes demonstrated statistically significant effects on the clinical decision for appendectomy. The presence of a positive McBurney sign had a major effect on the clinical decision compared to a negative sign (RRR, 47.73; 95% CI, 22.91–99.40). The presence of vomiting demonstrated a statistically significant effect on the clinical decision at the ‘maybe’ level of the attribute compared to the ‘no’ level (RRR, 2.31; 95% CI, 1.04–5.14). In addition, the decision to perform appendectomy was significantly influenced by the white blood cell count (WBCC) at a level ≥15 × 10^9^/L in comparison to a level ≥10 × 10^9^/L (RRR, 2.04; 95% CI, 1.02–2.30). No other clinical factor significantly influenced the decision to perform appendectomy.

### 3.4. Determinants of Preoperative Preparation for Appendectomy

A clinical examination, patient consent, and other procedures significantly affected the preoperative preparation for appendectomy. The patient history and a physical examination were the most common determinants in the preoperative preparation for appendectomy compared to the patient history and examination separately (RRR, 2.86; 95% CI, 1.66–4.94). Mandatory patient consent was also more common in preoperative preparation than optional consent (RRR, 1.93; 95% CI, 1.24–3.02).

## 4. Discussion

Our conjoint analysis demonstrated that the decision making to perform appendectomy was influenced by several factors, including the McBurney sign, vomiting, and WBCC. The preoperative preparation was mainly influenced by a clinical examination and patient consent. We observed wide confidence intervals for each clinical attribute, which indicated variation in clinical decision making.

### 4.1. Strengths and Limitations

To our knowledge, this is the first study to analyse the decision making to perform appendectomy in Kosovo and the region (Balkans), focusing on the clinical attributes that influence it. In addition, this study examined the effect of the clinical attributes that define the preoperative preparation for appendectomy. Using conjoint analysis, we quantified the impact of clinical attributes on the decision to perform appendectomy and on the preoperative preparation. The decision to perform appendectomy is minimally documented. Hence, our study provides a practical design and evidence in this domain. The small sample size was the main limitation of this study [13,20,21,22]. The small sample size limited the generalization of our findings and did not allow proper subgroup analysis; hence, we could not examine the effects of the project intervention. A small sample size can be feasible and functional [14]; nevertheless, it proved insufficient for performing a full analysis and drawing proper conclusions about the effect of the intervention. Another limitation is that pretesting may have failed to identify the McBurney sign as an absolute indicator, hence why it had such a strong effect on the analysis. Following a thorough procedure is, on the other hand, one of the biggest strengths of this study. A thorough consideration of the literature was performed to define and design the clinical attributes (and levels of these attributes) influencing the decision for appendectomy and the preoperative preparation for appendectomy, test the questionnaire before the final administration, and conduct a thorough follow-up of the steps in performing the conjoint analysis.

### 4.2. Interpretation, Context, and Implications 

Appendectomy has been recognized as a standard treatment for appendicitis for a long time. Nevertheless, the successful management of patients without surgery is evident in several studies [24,25]. One of the most recent studies suggests that female sex, appendiceal diameter, and the presence of an appendicolith are factors that result in increased odds of having to undergo an appendectomy compared to using antibiotic treatments [25].

The process of making a balanced judgement on whether to perform an appendectomy or not remains very challenging, mainly due to the variation in symptoms and signs in different patients, but also due to the lack of standardized care and organizational barriers [25]. That being said, there are several reasons why the insights derived from this study are of great importance. To begin with, they present an emerging need to recognize how a clinical decision for appendectomy is made. Secondly, the evidence-based practice should be implemented and adopted because it provides a standard with the potential to improve clinical outcomes at a cost-effective ratio.

The initial findings indicate a lack of standardization of clinical decision making and preoperative care related to appendectomies. Another round of repeating the same design later in the implementation phase is a recommended option, especially if conducted with a larger number of surgeons from other public hospitals. This would allow time for the effects of an intervention to take place. It would also allow for a post-intervention analysis with a comparison of the results with the baseline. Finally, a larger number of hospitals would mean a larger sample size, which could be helpful for the generalization of findings in the country context and beyond.

## 5. Conclusions

In conclusion, the clinical decisions and preparation for preoperative procedures in Kosovo are influenced by different factors and can vary among surgeons. We found that several factors influenced the decision to perform appendectomy and the practices for preoperative preparation. At times, these factors have little clinical importance as compared to other signs, and at other times, they are not relevanto the clinical decision. There is also wide variation in the effect of clinical attributes. Nevertheless, the small sample size limited our efforts to interpret the results. These findings could assist Kosovo in the design and implementation of future similar studies and in fostering quality improvement measures for addressing clinical decision making and the lack of process standardization in the delivery of surgical care. Most attribute-stated preference studies in healthcare are focused on high-income countries [13,26], and a very limited number of such studies focus on low-income and low-to-medium-income countries. Hence these study results may provide useful information and serve as a model for other low- to middle-income countries.

## Figures and Tables

**Table 1 ijerph-19-14027-t001:** Attributes and levels for clinical decision for appendectomy.

Attributes and Levels for Clinical Decision for Appendectomy
Attribute by levels
McBurney sign
Negative
Positive
Vomiting present
No
Maybe
Yes
Temperature
≥37.7 °C
≥38.5 °C
White blood cell count
≥10 × 10^9^/L
≥15 × 10^9^/L
Erythrocyte sedimentation in blood
0–20 mm/h
20 mm/h
>20 mm/h
C-reactive protein in the blood
>10 mg/L
>20 mg/L
Misleading causes typical for other diseases
Chronic diarrhoea
Constipation
Swelling in the affected area

**Table 2 ijerph-19-14027-t002:** Attributes and levels for preoperative preparation for appendectomy.

Attributes and Levels for Preoperative Preparation for Appendectomy
Attribute by levels
Clinical examination
Patient history
Patient history and physical examination
Patient examination
Clinical measures
Blood pressure measurement
Height measurement
Pulse measurement
Weight measurement
Comorbidities
Comorbidities are not mandatory to be checked
Verify present and specified comorbidities
Verify previous (not present) comorbidities
Diagnostic procedures
Lab findings
Abdominal ultrasound
CT scan
MRI
Other procedures
Colon lavage
Stomach lavage
Patient consent
Optional
Mandatory

## Data Availability

Not applicable.

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
