# Peer review of "Clinical Decision-Making for Appendectomy in Kosovo: A Conjoint Analysis"

_ijerph, 2022, doi:10.3390/ijerph192114027_

Round 1

Reviewer 1 Report

The authors have attempted conjoint analysis, one inherent research method in economics for clinical decision-making, which should be appreciated. There are only limited papers in this line, and many, like Daniel Kahneman, might object to clinical decision-making into some quantitative matrix. (Kahneman, 2012)

However, the present manuscript is a well-structured and well-written one. Unfortunately, it seems that the authors have not given importance to the basic tenants of conjoint analysis; the sample size reported is insufficient, especially for the choice-based analysis that is stated in the paper. Kindly see the reference, which clearly states the sample size requirements for conjoint analysis. (Orme, 2009)

There is no wonder why the results are not the same on subgroup analysis. I cannot understand how the authors can even attempt subgroup analysis with this insufficient sample size.

Kahneman D (2012) Thinking, Fast and Slow. London: Penguin.

Orme BK (2009) Sample Size Issues for Conjoint Analysis- Chapter 7. In: Getting Started With Conjoint Analysis: Strategies for Product Design and Pricing Research. 2nd edition. Madison, Wis: Research Pub Llc.

Author Response

Point 1: The authors have attempted conjoint analysis, one inherent research method in economics for clinical decision-making, which should be appreciated. There are only limited papers in this line, and many, like Daniel Kahneman, might object to clinical decision-making into some quantitative matrix. (Kahneman, 2012)
Response 1: Thank you for your comment. We agree that conjoint analysis is rarely used for the assessment of clinical decision-making. But since we have read Danishevski et al. (2008) we always thought that there is enormous potential for assessing clinical decision-making, particularly medical
practice variation. Variation in clinical decision-making is an important source of problems in the delivery of care, where inappropriate care is over-provided and needed care is under-provided. Studying this at the micro level, that is, the physician level is extremely useful as compared to studying at the regional level because it provides direct insight into what is happening in clinical decision-making. Which is easier to understand (causes for variation) and more helpful in changing existing practice. Danishevski, K., McKee, M., Sassi, F., & Maltcev, V. (2008). The decision to perform Caesarean section in Russia. Int J Qual Health Care, 20(2), 88-94.https://doi.org/10.1093/intqhc/mzm070

Reference to manuscript: Lines 82-89 of the manuscript

Point 2: Unfortunately, it seems that the authors have not given importance to the basic tenants of conjoint analysis; the sample size reported is insufficient, especially for the choice-based analysis that is stated in the paper. Kindly see the reference, which clearly states the sample size requirements for conjoint analysis. (Orme, 2009)
Response 2: Thank you for this comment. We were aware of the reference as well as this limitation of our study. Nevertheless, we decided to pursue publication for the following reasons:
1. This study was designed with care and attention to every detail, respecting the latest developments (we were aware) in conjoint analysis design. We spent over a year doing this. All steps are described in the manuscript. As such, it represents a valuable resource for other researchers who want to pursue similar analysis.
2. The sample included in the study represents 91.7% of the total population of surgeons in both hospitals. So it is representative of the population under study. As Orme 2009 states in reference, you have shared with us: ” Conjoint studies may be used for large or small populations. We can use conjoint analysis for even the smallest of populations, provided we interview enough respondents to
represent the population adequately.” Which is our case. We have represented the population adequately.
3. Despite sample size limitation, the analysis is largely valuable as it can highlight and expose the clinical variation and particular clinical factors that influence that and the direction of such influence.

4. For similar reasons and justification Danishevski et al. have performed conjoint analysis for Cesarean section with a lower than 100 sample size. 5. Bridges et al. have performed the conjoint analysis with a similar sample (like our study) and highlighted that conjoint analysis with such a sample “can be functional and feasible”. 6. A review of the conjoint analysis found that many papers limit their analysis to small sample sizes (up to 13 participants in one case). Danishevski, K., McKee, M., Sassi, F., & Maltcev, V. (2008). The decision to perform Caesarean section in Russia. Int J Qual Health Care, 20(2), 88-94. https://doi.org/10.1093/intqhc/mzm070 https://pubmed.ncbi.nlm.nih.gov/22047535/ https://www.ncbi.nlm.nih.gov/pmc/articles/ PMC3529196/ https://pubmed.ncbi.nlm.nih.gov/23110423/ https://pubmed.ncbi.nlm.nih.gov/22273432/ To address this comment we have added additional explanations on the issue in specific sections of the manuscript.

Reference to manuscript: Methods and discussion section of the manuscript

Point 3: There is no wonder why the results are not the same on subgroup analysis. I cannot understand how the authors can even attempt subgroup analysis with this insufficient sample size.

Response 3: Thank you for this comment. We agree with your observation. Out of enthusiasm to show the differences between intervention and non-intervention, we had planned and pursued this. To address your comments we have taken out subgroup analysis from the manuscript.

Reference to manuscript: Throughout the manuscript.

Reviewer 2 Report

The research topic is not very relevant for such an international journal. The methodology, while interesting, is explained in a very unstructured and unclear way.  The methodology is one of the points for improvement. On the other hand, the results could be improved and the conclusions are very scarce. The bibliographical references are quite outdated; only 4 of them are less than 5 years old. The methodological approach is good, but the subject matter is not interesting. Neither are the results. 

Author Response

Point 1: The research topic is not very relevant for such an international journal.

Response 1: We respectfully disagree with the comment. First of all, the paper links well with the call, and secondly, the importance of variation in clinical decision making, which is the initial moment which then reflects in hospital, regional and national variations in care delivery are recognised as some of the most important issues in the delivery of care as highlighted by reputable journals and researchers. Largely because they cause over and underuse of care and poor quality of care, no matter which country you live in. 

Berwick, D. M. (2017). Avoiding overuse—the next quality frontier. The Lancet, 390(10090), 102-104. https://doi.org/10.1016/s0140-6736(16)32570-3

Saini, V., Brownlee, S., Elshaug, A. G., Glasziou, P., & Heath, I. (2017). Addressing overuse and underuse around the world. The Lancet, 390(10090), 105-107. https://doi.org/10.1016/s0140-6736(16)32573-9

Saini, V., Garcia-Armesto, S., Klemperer, D., Paris, V., Elshaug, A. G., Brownlee, S., Ioannidis, J. P. A., & Fisher, E. S. (2017). Drivers of poor medical care. Lancet, 390(10090), 178-190. https://doi.org/10.1016/s0140-6736(16)30947-3

We have highlighted this in the manuscript.

Reference to manuscript: Lines 82-89 of the manuscript

Point 2: The methodology, while interesting, is explained in a very unstructured and unclear way.  The methodology is one of the points for improvement.

Response 2: Thank you for your suggestion. We have revised the methodology section and introduced some subheadings. We have also extended some parts to make it more clear.

Reference to manuscript: Methods section

 Point 3: On the other hand, the results could be improved and the conclusions are very scarce. 

Response 3: Thank you for your comment. We have tried to improve this to the extent possible. We have also included subheadings to make the read easier.

Reference to manuscript: Results and conclusion section

 Point 4: The bibliographical references are quite outdated; only 4 of them are less than 5 years old.

Response 4: Thank you for your comment. We have tried to update references to the extent possible. 

Reference to manuscript: Throughout the manuscript

Point 5: The methodological approach is good, but the subject matter is not interesting

Response 5: Thank you for your comment. We respectfully disagree with the second part of the comment. See the response to the previous comment.

Point 6: Neither are the results.

Response 6: We respectfully disagree. This is the first conjoint analysis to examine appendectomy decision-making. Results have limitations due to sample size. But they are more than relevant for the country, and we think they may be relevant to a wider readership interested in the domain of clinical decision-making variation and quality of care.

Round 2

Reviewer 1 Report

Thank you, the manuscript is substantially improved with the revisions. However, the basic issue of low sample size still prevails. My concern is not only about the representativeness but mainly about the validity of the analysis method. When examining 17 clinical scenarios among 22 participants, the vanity of the methods goes down.

However, I appreciate the innovative conjoint analysis approach in the clinical domain. I feel the manuscript can be structured as a short communication or correspondence after redrafting it as per the good research practices for conjoint analysis as advocated by Bridges et al.(Bridges JFP, Hauber AB, Marshall D, et al. (2011) Conjoint analysis applications in health--a checklist: a report of the ISPOR Good Research Practices for Conjoint Analysis Task Force. Value in Health: The Journal of the International Society for Pharmacoeconomics and Outcomes Research 14(4): 403–413. DOI: 10.1016/j.jval.2010.11.013.)

Author Response

Dear Reviewer 1,

We have decided to publish the manuscript as a communication. We have tried to re-organize the draft so it matches editorial and both reviewers comments as well as the communication format of the journal (based on similar papers published at IJERPH).

Reviewer 2 Report

Significant improvements have been made to the structure and quality of the article. But the way in which they extrapolate the results and draw conclusions with only 22 patients is not adequate. I suggest you expand the sample, complete the study, and publish it later.

Author Response

Dear Reviewer 2,

We have decided to publish the manuscript as a communication. We have tried to re-organize the draft so it matches editorial and both reviewers comments as well as the communication format of the journal (based on similar papers published at IJERPH).